# LEARNING WORLD GRAPH DECOMPOSITIONS TO ACCELERATE REINFORCEMENT LEARNING

## ABSTRACT

Efficiently learning to solve tasks in complex environments is a key challenge for reinforcement learning (RL) agents. We propose to decompose a complex environment using a task-agnostic *world graphs*, an abstraction that accelerates learning by enabling agents to focus exploration on a subspace of the environment. The nodes of a world graph are important *waypoint* states and edges represent feasible traversals between them. Our framework has two learning phases: 1) identifying world graph nodes and edges by training a binary recurrent variational autoencoder (VAE) on trajectory data and 2) a hierarchical RL framework that leverages structural and connectivity knowledge from the learned world graph to bias exploration towards task-relevant waypoints and regions. We thoroughly evaluate our approach on a suite of challenging maze tasks and show that using world graphs significantly accelerates RL, achieving higher reward and faster learning.

## 1 INTRODUCTION

Many real-world applications, e.g., self-driving cars and in-home robotics, require an autonomous agent to execute different tasks within a single environment that features, e.g. high-dimensional state space, complex world dynamics or structured layouts. In these settings, model-free reinforcement learning (RL) agents often struggle to learn efficiently, requiring a large amount of experience collections to converge to optimal behaviors. Intuitively, an agent could learn more efficiently by focusing its exploration in *task-relevant regions*, if it has knowledge of the high-level structure of the environment.

We propose a method to 1) learn and 2) use an environment decomposition in the form of a *world graph*, a *task-agnostic* abstraction. World graph nodes are *waypoint* states, a set of salient states that can summarize agent trajectories and provide meaningful starting points for efficient exploration (Chatzigiorgaki & Skodras, 2009; Jayaraman et al., 2018; Ghosh et al., 2018). The directed and weighted world graph edges characterize feasible traversals among the waypoints. To leverage the world graph, we model hierarchical RL (HRL) agents where a high-level policy chooses a waypoint state as a goal to guide exploration towards task-relevant regions, and a low-level policy strives to reach the chosen goals.

Our framework consists of two phases. In the task-agnostic phase, we obtain world graphs by training a recurrent variational auto-encoder (VAE) (Chung et al., 2015; Gregor et al., 2015; Kingma & Welling, 2013) with binary latent variables (Nalisnick & Smyth, 2016) over trajectories collected using a random walk policy (Ha & Schmidhuber, 2018) and a curiosity-driven goal-conditioned policy (Ghosh et al., 2018; Nair et al., 2018). World graph nodes are states that are most frequently selected by the binary latent variables, while edges are inferred from empirical transition statistics between neighboring waypoints. In the task-specific phase, taking advantage of the learned world graph for structured exploration, we efficiently train an HRL model (Taylor & Stone, 2009).

In summary, our main contributions are:

- A task-agnostic unsupervised approach to learn world graphs, using a recurrent VAE with binary latent variables and a curiosity-driven goal-conditioned policy.
- An HRL scheme for the task-specific phase that features multi-goal selection (Wide-then-Narrow) and navigation via world graph traversal.

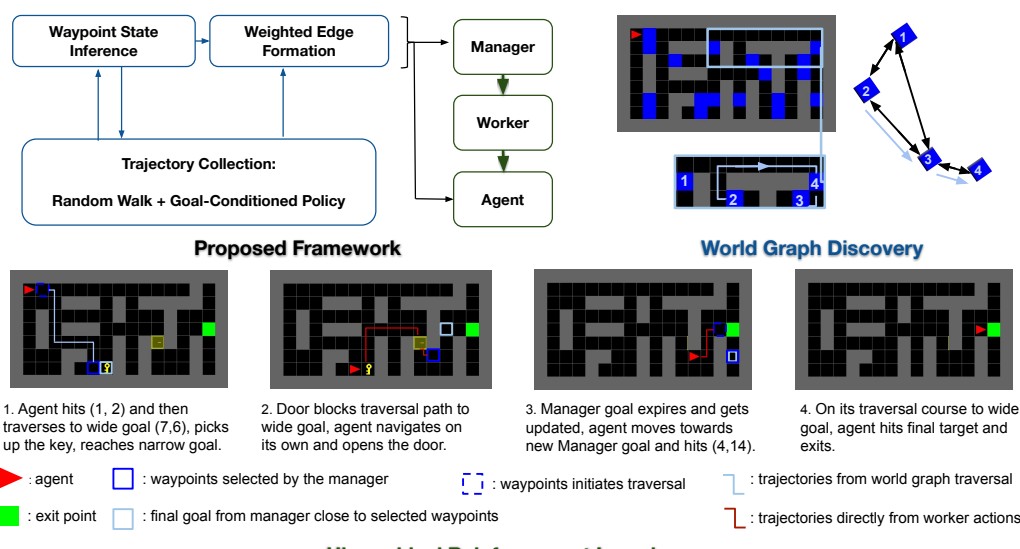

Figure 1: Top Left: overall pipeline of our 2-phase framework. Top Right (*world graph discovery*): a subgraph exemplifies traversal between waypoint states (in blue), see Section 3 for more details. Bottom (*Hierarhical RL*): an example rollout from our proposed HRL policy with Wide-then-Narrow Manager instructions and world graph traversals, solving a challenging *Door-Key* task, see Section 4 for more details.

- Empirical evaluations on multiple tasks in complex 2D grid worlds to validate that our framework produces descriptive world graphs and significantly improves both sample efficiency and final performance on these tasks over baselines, especially thanks to transfer learning from the unsupervised phase and world graph traversal.

## 2 RELATED WORK

An understanding of the environment and its dynamics is essential for effective planning and control in model-based RL. For example, a robotics agent often locates or navigates by interpreting a map (Lowry et al., 2015; Thrun, 1998; Angeli et al., 2008). Our exploration strategy draws inspiration from active localization, where robots are actively guided to investigate unfamiliar regions (Fox et al., 1998; Li et al., 2016). Besides mapping, recent works (Azar et al., 2019; Ha & Schmidhuber, 2018; Guo et al., 2018) learn to represent the world with generative latent states (Tian & Gong, 2017; Haarnoja et al., 2018; Racanière et al., 2017). If the latent dynamics are also extrapolated, the latent states can assist planning (Mnih et al., 2016a; Hafner et al., 2018) or model-based RL (Gregor & Besse, 2018; Kaiser et al., 2019).

While also aiming to model the world, we approach this as abstracting both the structure and dynamics of the environment in a graph representation, where nodes are states from the environment and edges encode actionable efficient transitions between nodes. Existing works (Metzen, 2013; Mannor et al., 2004; Eysenbach et al., 2019; Entezari et al., 2010) have shown benefits of such graph abstractions but typically select nodes only subject to a good coverage the observed state space. Instead, we identify a parsimonious subset of states that can summarize trajectories and provide more useful intermediate landmarks, i.e. waypoints, for navigating complex environments.

Our method for estimating waypoint states can be viewed as performing automatic (sub)goal discovery. Subgoal and subpolicy learning are two major approaches to identify a set of temporally-extended actions, "skills", that allow agents to efficiently learn to solve complex tasks. Subpolicy learning identifies policies useful to solve RL tasks, such as option-based methods (Daniel et al., 2016; Bacon et al., 2017) and subtask segmentations (Pertsch et al., 2019; Kipf et al., 2018). Subgoal learning, on the other hand, identifies "important states" to reach (Şimşek et al., 2005).

Previous works consider various definitions of "important" states: frequently visited states during successful task completions (Digney, 1998; McGovern & Barto, 2001), states introducing the most novel information (Goyal et al., 2019), bottleneck states connecting densely-populated regions (Chen

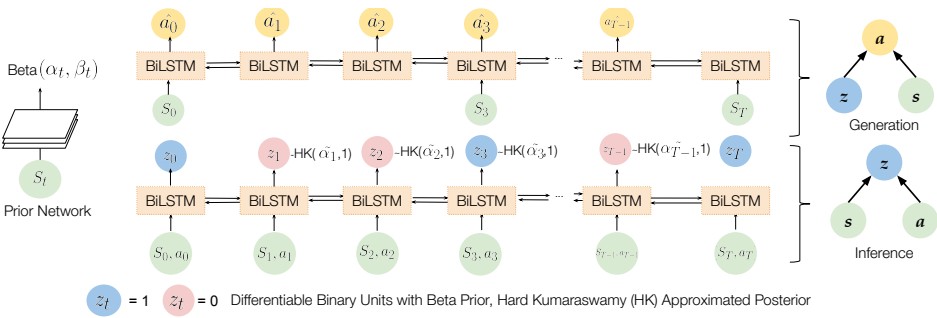

Figure 2: Our recurrent latent model with differentiable binary latent units to identify waypoint states. A prior network (left) learns the state-conditioned prior in Beta distribution, $p_\psi(z_t|s_t)=\text{Beta}(\alpha_t, \beta_t)$. An inference encoder learns an approximate posterior in HardKuma distribution inferred from the state-action sequence input, $q_\phi(z_t|\boldsymbol{a}, \boldsymbol{z})=\text{HardKuma}(\tilde{\alpha}_t, 1)$. A generation network $p_\theta$ reconstructs $\boldsymbol{a}$ from $\{s_t|z_t=1\}$.

et al., 2007; Şimşek et al., 2005), or environment-specific heuristics (Ecoffet et al., 2019). Our work draws intuition from unsupervised temporal segmentation (Chatzigiorgaki & Skodras, 2009; Jayaraman et al., 2018) and imitation learning (Abbeel & Ng, 2004; Hussein et al., 2017). We define "important" states (waypoints) as the most critical states in recovering action sequences generated by some agent, which indicates that these states contain the richest information about the executed policy (Azar et al., 2019).

## 3 LEARNING WORLD GRAPHS

We propose a method for learning a *world graph* $\mathcal{G}_w$, a task-agnostic abstraction of an environment that captures its high-level structure and dynamics. In this work, the primary use of world graphs is to accelerate reinforcement learning of downstream tasks. The nodes of $\mathcal{G}_w$, denoted by a set of *waypoints* states $s_p \in \mathcal{V}_p$, are generically "important" for accomplishing tasks within the environment, and therefore useful as starting points for exploration. Our method identifies such waypoint states from interactions with the environment. In addition, we embed feasible transitions between nearby waypoint states as the edges of $\mathcal{G}_w$.

In this work, we define important states in the context of *learning* $\mathcal{G}_w$ (see Section 2 for alternative definitions). That is, we wish to discover a small set of states that, when used as world graph nodes, concisely summarize the structure and dynamics of the environment. Below, we describe 1) how to collect state-action trajectories and an unsupervised learning objective to identify world graph nodes, and 2) how the graph's edges (i.e., how to transition between nodes) are formed from trajectories.

### 3.1 WAYPOINT STATE IDENTIFICATION

The structure and dynamics of an environment are implicit in the state-action trajectories observed during exploration. To identify world graph nodes from such data, we train a recurrent variational autoencoder (VAE) that, given a sequence of state-action pairs, identifies a subset of the states in the sequence from which the full action sequence can be reconstructed (Figure 2). In particular, the VAE infers *binary latent variables* that controls whether each state in the sequence is used by the generative decoder, i.e., whether a state is "important" or not.

**Binary Latent VAE** The VAE consists of an inference, a generative and a prior network. These are structured as follows: the input to the inference network $q_\phi$ is a trajectory of state-action pairs observed from the environment $\tau=\{(s_t, a_t)\}_{t=0}^T$, with $\boldsymbol{s}=\{s_t\}_{t=0}^T$ and $\boldsymbol{a}=\{a_t\}_{t=0}^T$ denoting the state and action sequences respectively. The output of the inference network is the approximated posterior over a sequence $\boldsymbol{z}=\{z_t\}_{t=0}^T$ of binary latent variables, denoted as $q_\phi(\boldsymbol{z}|\boldsymbol{a}, \boldsymbol{s})$. The generative network $p_\theta$ computes a distribution over the full action sequence $\boldsymbol{a}$ using the *masked* state sequence, where $s_t$ is masked if $z_t=0$ (we fix $z_0=z_T=1$ during training), denoted as $p_\theta(\boldsymbol{a}|\boldsymbol{s}, \boldsymbol{z})$.

Finally, a *state-conditioned* $p_\psi(z_t|s_t)$ given by the prior network $p_\psi$ for each $s_t$ encodes the empirical average probability that state $s_t$ is activated for reconstruction. This choice encourages inference to select within a consistent subset of states for use in action reconstruction. In particular, *the waypoint*

---

**Algorithm 1:** Identifying waypoint states $\mathcal{V}_p$ and learning a goal-conditioned policy $\pi_g$

---

**Result:** Waypoint states $\mathcal{V}_p$ and a goal-conditioned policy $\pi_g$
Initialize network parameters for the recurrent variational inference model $V$
Initialize network parameters for the goal-conditioned policy $\pi_g$
Initialize $\mathcal{V}_p$ with the initial position of the agent, i.e. $\mathcal{V}_p = \{s_0 = (1,1)\}$
**while** *VAE reconstruction error has not converged* **do**

    **for** $n \leftarrow 1$ **to** $N$ **do**
        Sample random waypoint $s_p \in \mathcal{V}_p$
        Navigate agent to $s_p$ and perform $T$-step rollout using a *randow walk* policy:
        $\tau_n^r \leftarrow \{(s_0 = s_p, a_0), ..., (s_T, a_T)\}$
        $g_n \leftarrow s_T$
        Navigate agent to $s_p$ and perform $T$-step rollout using $\pi_g$ with goal $g_n$:
        $\tau_n^\pi \leftarrow \{(s_0 = s_p, a_0), ..., (s_T, a_T)\}_{a_t \sim \pi_g(\cdot|s_t, g_n)}$
        Re-label $\pi_g$ rewards with action reconstruction error as curiosity bonus:
        $r_n^\pi \leftarrow \{\mathbb{1}_{s_{t+1}=g_n} - \lambda \cdot p_\theta(a_t|\boldsymbol{s}, \boldsymbol{z})\}_{t=0}^T$
    **end**
    Perform policy gradient update of $\pi_g$ using $\tau^\pi$ and $r^\pi$
    Update $V$ using $\tau^r$ and $\tau^\pi$
    Update $\mathcal{V}_p$ as set of states with largest prior mean $\frac{\alpha_s}{\alpha_s + \beta_s}$.
**end**

---

*states $\mathcal{V}_p$ are chosen as the states with the largest prior means* and during training, once every few iterations, $\mathcal{V}_p$ is updated based on the current prior network.

**Objective** Formally, we optimize the VAE using the following evidence lower bound (ELBO):

$$\text{ELBO} = \mathbb{E}_{q_\phi(\boldsymbol{z}|\boldsymbol{a},\boldsymbol{s})} \left[\log p_\theta(\boldsymbol{a}|\boldsymbol{s}, \boldsymbol{z})\right] - D_{\text{KL}}\left(q_\phi(\boldsymbol{z}|\boldsymbol{a}, \boldsymbol{s})|p_\psi(\boldsymbol{z}|\boldsymbol{s})\right). \tag{1}$$

To ensure differentiablity, we apply a continuous relaxation over the discrete $z_t$. We use the Beta distribution $p_\psi(z_t) = \text{Beta}(\alpha_t, \beta_t)$ for the prior and the Hard Kumaraswamy distribution $q_\psi(z_t|\boldsymbol{a}, \boldsymbol{z}) = \text{HardKuma}(\tilde{\alpha}_t, \tilde{\beta}_t)$ for the approximate posterior, which resembles the Beta distribution but is outside the exponential family (Bastings et al., 2019). This choice allows us to sample 0s and 1s without sacrificing differentiability, accomplished via the stretch-and-rectify procedure (Bastings et al., 2019; Louizos et al., 2017) and the reparametrization trick (Kingma & Welling, 2013). Lastly, to prevent the trivial solution of using all states for reconstruction, we use a secondary objective $\mathcal{L}_0$ to regularize the $L_0$ norm of $\boldsymbol{z}$ at a targeted value $\mu_0$ (Louizos et al., 2017; Bastings et al., 2019), the desired number of selected states out of $T$ steps, e.g. for when $T = 25$, we set $\mu_0 = 5$, meaning ideally 5 out of 25 states are activated for action reconstruction. Another term $\mathcal{L}_T$ to encourage temporal separation between selected states by targeting the number of 0/1 switches among $\boldsymbol{z}$ at $2\mu_0$:

$$\mathcal{L}_0 = \left\|\mathbb{E}_{q_\phi(\boldsymbol{z}|\boldsymbol{s},\boldsymbol{a})}[\|\boldsymbol{z}\|_0] - \mu_0\right\|^2, \quad \mathcal{L}_T = \left\|\mathbb{E}_{q_\phi(\boldsymbol{z}|\boldsymbol{s},\boldsymbol{a})}\left[\sum_{t=0}^T \mathbb{1}[z_t \neq z_{t+1}]\right] - 2\mu_0\right\|^2. \tag{2}$$

See Appendix A for details on training the VAE with binary $z_t$, including integration of the Hard Kumaraswamy distribution and how to regularize the statistics of $\boldsymbol{z}$.

## 3.2 EXPLORATION FOR WORLD GRAPH DISCOVERY

Naturally, the latent structure learned by the VAE depends on the trajectories used to train it. Hence, collecting a rich set of trajectories is crucial. Here, we propose a strategy to bootstrap a useful set of trajectories by alternately exploring the environment based on the current iteration's $\mathcal{V}_p$ and updating the VAE and $\mathcal{V}_p$, repeating this cycle until the action reconstruction accuracy plateaus (Algorithm 1).

During exploration, we use action replay to navigate the agent to a state drawn from the current iteration's $\mathcal{V}_p$. Although resetting via action replay assumes our underneath environment to be deterministic, in cases where this resetting strategy is infeasible, it may be modified so long as to allow the exploration starting points to expand as the agent discovers more of its environment. For

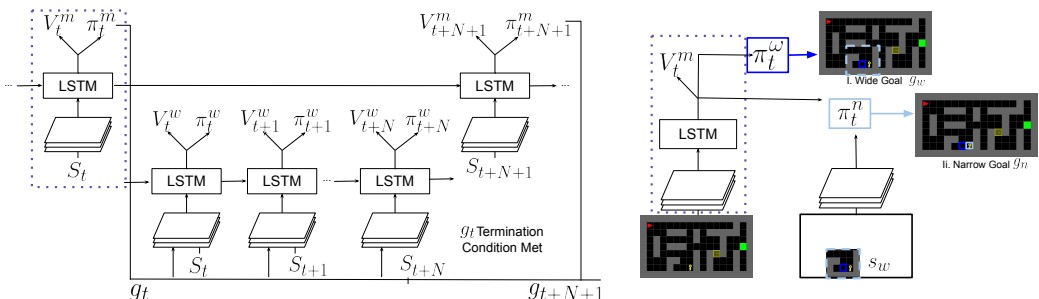

Figure 3: Left: a standard Feudal Network. Right: using Wide-then-Narrow goals. The Manager first outputs a waypoint state as the wide goal $g^w$, then attends to a closer-up area around $g^w$ to narrow down the final goal $g^n$.

each such starting point, we collect two rollouts. In the first rollout, we perform a random walk to explore the nearby region. In the second rollout, we perform actions using a goal-conditioned policy $\pi_g$ (GCP), setting the final state reached by the random walk as the goal. Both rollouts are used for trianing the VAE and the latter is also used for training $\pi_g$.

GCP provides a venue to integrate intrinsic motivation, such as curiosity (Burda et al., 2018; Achiam & Sastry, 2017; Pathak et al., 2017; Azar et al., 2019) to generate more diverse rollouts. Specifically, we use the action reconstruction error of the VAE as an intrinsic reward signal when training $\pi_g$. This choice of curioisty also prevents the VAE from collapsing to the simple behaviors of a vanilla $\pi_g$.

### 3.3 EDGE FORMATION

The final stage is to construct the edges of $\mathcal{G}_w$, which should ideally capture the environment dynamics, i.e. how to transition between waypoint states. Once VAE training is complete and $\mathcal{V}_p$ is fixed, we collect random walk rollouts from each of the waypoints $s_p \in \mathcal{V}_p$ to estimate the underlying adjacency matrix (Biggs, 1993). More precisely, we claim a directed edge $s_p \rightarrow s_q$ if there exists a random walk trajectory from $s_p$ to $s_q$ that does not intersect a third waypoint. We also consider paths taken by $\pi_g$ (starting at $s_p$ and setting $s_q$ as the goal) and keep the shortest observed path from $s_p$ to $s_q$ as a world graph edge transition. We use the action sequence length of the edge transition between adjacent waypoints as the *weight* of the edge. As shown experimentally, a key benefit of our approach is the ability to plan over $\mathcal{G}_w$. To navigate from one waypoint to another, we can use dynamic programming (Sutton, 1998; Feng et al., 2004) to output the optimal traversal of the graph.

## 4 ACCELERATING REINFORCEMENT LEARNING WITH WORLD GRAPHS

World graphs present a high-level, task-agnostic abstraction of the environment through waypoints and feasible transition routes between them. A key example of world graph applications for task-specific RL is *structured exploration*: instead of exploring the entire environment, RL agents can use world graphs to quickly identify task-relevant regions and bias low-level exploration to these regions. Our framework to leverage world graphs for structured exploration consists of two parts:

1. Hierarchical RL wherein the high-level policy selects subgoals from $\mathcal{V}_p$.
2. Traversals using world graph edges.

### 4.1 HIERARCHICAL RL OVER WORLD GRAPHS

Formally, an RL agent learning to solve a task is formulated as a Markov Decision Process: at time $t$, the agent is in a state $s_t$, executes an action $a_t$ via a policy $\pi(a_t|s_t)$ and receives a rewards $r_t$. The agent's goal is to maximize its cumulative expected return $R = \mathbb{E}_{(s_t,a_t) \sim \pi, p, p_0} \left[ \sum_{t \geq 0} \gamma^t r_t \right]$, where $p(s_{t+1}|s_t, a_t), p_0(s_0)$ are the transition and initial state distributions.

To incorporate world graphs with RL, we use a hierarchical approach based on the Feudal Network (FN) (Dayan & Hinton, 1993; Vezhnevets et al., 2017), depicted in Figure 3. A standard FN

| Task | Task Description | Environment Characteristics |
|------|-----------------|----------------------------|
| *MultiGoal* | Collect randomly spawned balls, each ball gives +1 reward. To end an episode, the agent has to exit at a designated point. | Balls are located randomly, dense reward. |
| *MultiGoal-Sparse* | Agents receive a single reward $r \leq 1$ proportional to the number of balls collected upon exiting. | Balls are located randomly, sparse reward. |
| *MultiGoal-Stochastic* | Spawn lava blocks at random locations each time step that immediately terminates the episode if stepped on. | *Stochastic* environment. Multiple objects: lava and balls are randomly located, dense reward. |
| *Door-Key* | Agent has to pick up a key to open a door (reward +1) and reach the exit point on the other side (reward +1). | Walls, door and key are located randomly. Agents have additional actions: pick and toggle. |

Table 1: An overview of tasks used to evaluate the benefit of using world graphs. Visualizations can be found in Appendix D.

decomposes the policy of the agent into two separate policies that receive distinct streams of reward: a high-level policy ("Manager") learns to propose subgoals; a low-level policy ("Worker") receives subgoals from the Manager as inputs and is rewarded for taking actions in the environment that reach the subgoals. The Manager receives the environment reward defined by the task and therefore must learn to emit subgoals that lead to task completion. The Manager and Worker do not share weights and operate at different temporal resolutions: the Manager only outputs a new subgoal if either the Worker reaches the chosen one or a subgoal horizon $c$ is exceeded.

For all our experiments, policies are trained using advantage actor-critic (A2C), an on-policy RL algorithm (Wu & Tian, 2016; Pane et al., 2016; Mnih et al., 2016b). To ease optimization, the feature extraction layers of the Manager and Worker that encode $s_t$ are initialized with the corresponding layers from $\pi_g$, the GCP learned during world graph discovery phase. More details are in Appendix B.

### 4.2 Wide-then-Narrow Goals and World Graphs

To incorporate the world graph, we introduce a Manager policy that factorizes subgoal selection as follows: a *wide* policy $\pi^w(g_t^w|s_t)$ selects a waypoint state as the *wide* goal $g^w \in \mathcal{V}_p$, and a *narrow* policy $\pi^n(g_t^n|s_t, g_t^w)$ selects a state within a local neighborhood of $g_t^w$, i.e. its $\epsilon$-net (Mahadevan & Maggioni, 2007), as the *narrow* goal $g^n \in \{s : \mathcal{D}(s, g_t^w) \leq \epsilon\}$. The Worker policy $\pi^{\text{worker}}(a_t|s_t, g_t^n, g_t^w)$ chooses the action taken by the agent given the current state and the wide and narrow goals from the Manager. A visual illustration is in Figure 4 and training details in Appendix C.2.

### 4.3 World Graph Traversal

The wide-then-narrow subgoal format simplifies the search space for the Manager policy. Using waypoints as wide goals also makes it possible to leverage the edges of the world graph for planning and executing the planned traversals. This process breaks down as follows:

1. **When to Traverse:** When the agent encounters a waypoint state $s_t \in \mathcal{V}_p$, a "traversal" is initiated if $s_t$ has a feasible connection in $\mathcal{G}_w$ to the active wide goal $g_t^w$.

2. **Planning:** Upon triggering a traversal, the optimal traversal route from the initiating state to $g_t^w$ is estimated from the $\mathcal{G}_w$ edge weights using classic dynamic programming planning (Sutton, 1998; Feng et al., 2004). This yields a sequence of intermediate waypoint states.

3. **Execution:** Execution of graph traversals depends on the nature of the environment. If deterministic, the agent simply follows the action sequences given by the edges of the traversal. Otherwise, the agent uses the pretrained GCP $\pi_g$ to sequentially reach each of the intermediate waypoint states along the traversal (we fine-tune $\pi_g$ in parallel where applicable). If the agent fails to reach the next waypoint state within a certain time limit, it stops its current pursuit and a new $(g^w, g^n)$ pair is received from the Manager.

World graph traversal allows the Manager to assign task-relevant wide goals $g^w$ that can be far away from the agent yet still reachable, which consequentially accelerates learning by focusing exploration around the task-relevant region near $g^w$.

## 5 Experimental Validation

We now assess each component of our framework on a set of challenging 2D grid worlds. Our ablation studies demonstrate the following benefits of our framework:

| Task | Size | A2C | FN $+\pi_g$ init | Ours | | |
|------|------|-----|------------------|------|--|--|
| | | | | $+\pi_g$-**init** | $+ \mathcal{G}_w$-**traversal** | $+ \pi_g$-**init** $+ \mathcal{G}_w$-**traversal** |
| *MultiGoal* | Small | 2.04±0.05 | 2.93±0.74 | **5.25±0.13** | 3.92±0.22 | 5.05±0.03 |
| | Medium | - | - | **5.15±0.11** | 2.56±0.09 | 3.00±0.90 |
| | Larger | - | - | - | 2.18±0.12 | **2.72±0.59** |
| *MultiGoal-Sparse* | Small | - | - | 0.39±0.09 | 0.24±0.04 | **0.42±0.07** |
| | Medium | - | - | - | 0.20±0.04 | **0.25±0.03** |
| | Larger | - | - | - | 0.16±0.22 | **0.26±0.11** |
| *MultiGoal-Stochastic* | Small | 1.38±1.20 | 1.93±0.16 | **3.06±0.31** | - | 2.92±0.45 |
| | Medium | - | - | 2.99±0.12 | 2.42±0.24 | **2.64±0.14** |
| | Larger | - | - | - | - | **0.60±0.12** |
| *Door-Key* | Small | - | - | **0.99±0.00** | 0.37±0.15 | 0.92±0.02 |
| | Medium | - | - | 0.56±0.02 | - | **0.76±0.06** |
| | Larger | - | - | - | - | **0.26±0.19** |

Table 2: On a variety of tasks and environment setups, we evaluate RL models trained with GCP $\pi_g$ initialization, with $\mathcal{G}_w$ world graph travresal, and with both. All models on the right are equipped with WN. Left are baselines for additional comparison. We report final rewards for *MultiGoal* tasks and success rates for *Door-Key* are reported. If no result reported, the agent failed to solve the task.

| Waypoint type | *MultiGoal* | *MultiGoal-Sparse* | *MultiGoal-Stochastic* | *Door-Key* |
|---------------|-------------|--------------------|-----------------------|-----------|
| Learned | **2.72**±0.59 | **0.26**±0.11 | **0.60**±0.12 | **0.26**±0.19 |
| Random | 2.30±0.49 | 0.19±0.11 | 0.41±0.25 | **0.27**±0.40 |

Table 3: Comparing learned $\mathcal{V}_p$ versus random $\mathcal{V}_{\mathrm{rand}}$ as wide subgoals on large mazes, all trained with $\pi_g$ initialization and graph traversal. $\mathcal{V}_p$ generally is superior in terms of performance and consistency. We report final rewards for *MultiGoal* tasks and success rates for *Door-Key* are reported.

- It improves sample efficiency and performance over the baseline HRL model.
- It benefits tasks varying in enviroment scale, task type, reward structure, and stochasticity.
- The identified waypoints provide superior world representations for solving downstream tasks, as compared to graphs using randomly selected states as nodes.

Implementation details, snippets of the tasks and mazes are in Appendix C-D.

## 5.1 ABLATION STUDIES ON 2D GRID WORLDS

For our ablation studies, we construct 2D grid worlds of increasing sizes (small, medium and large) along with challenging tasks with different reward structures, levels of stochasticity and logic (summarized in Table 1). In all tasks, every action taken by the agent receives a negative reward penalty. We follow a rigorous evaluation protocol (Wu et al., 2017; Ostrovski et al., 2017; Henderson et al., 2018): each experiment is repeated with 3 training seeds. 10 additional validation seeds are used to pick the model with the best reward performance. This model is then tested on 100 testing seeds. We report mean reward and standard deviation.

We ablate each of the following components in our framework and compare against non-hierarchical (A2C) and hierarchical baselines (FN):

1. initializing the feature extraction layers of the Manager and Worker from $\pi_g$,
2. applying Wide-then-Narrow Manager (WN) goal instruction, and
3. allowing the Worker to traverse along $\mathcal{G}_w$.

Results are shown in Table 2. In sum, each component improves performance over the baselines.

**Wide and narrow goals**  Using two goal types is a highly effective way to structure the Manager instructions and enables the Worker to differentiate the transition and local task-solving phases. We note that for small *MultiGoal*, agents do not benefit much from $\mathcal{G}_w$ traversal: it can rely solely on the guidance from WN goals to master both phases. However with increasing maze size, the Worker struggles to master traversals on its own and thus fails solving the tasks.

**World Graph Traversal**  As conjectured in Section 4.3, the performance gain of our framework can be explained by the larger range and more targeted exploration strategy. In addition, the Worker

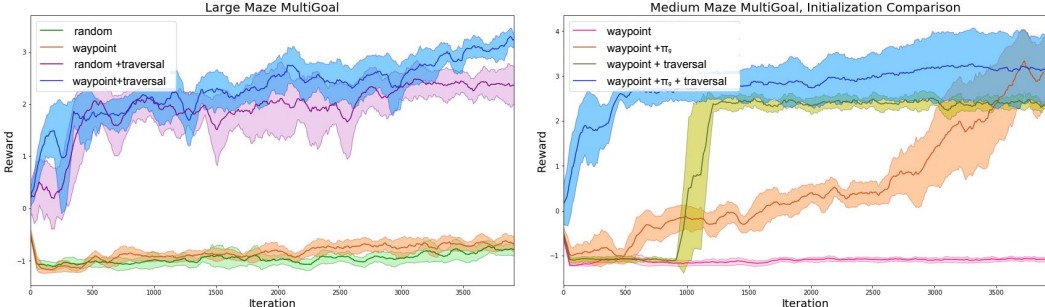

Figure 4: Validation performance during training (mean and standard-deviation of reward, 3 seeds) for *MultiGoal*. Left: Comparing $\mathcal{V}_p$ and $\mathcal{V}_{\mathrm{rand}}$, with or without traversal, all models use WN and $\pi_g$ initialization. We see that 1) traversal speeds up convergence, 2) $\mathcal{V}_{\mathrm{rand}}$ gives higher variance and slightly worse performance than $\mathcal{V}_p$. Right: comparing with or without $\pi_g$ initialization on $\mathcal{V}_p$, all models use WN. We see that initializing the task-specific phase with the task-agnostic goal-conditioned policy significantly boosts learning.

does not have to learn long distance transitions with the aid of $\mathcal{G}_w$ traversals. Figure 4 confirms that $\mathcal{G}_w$ traversal speeds up convergence and its effect becomes more evident with larger mazes. Note that the graph learning stage only need 2.4K iterations to converge. Even when taking these additional environment interactions into account, $\mathcal{G}_w$ traversal still exhibits superior sample efficiency, not to mention that the graph is shared among all tasks. Moreover, solving *Door-Key* involves a complex combination of sub-tasks: find and pick up the key, reach and open the door and finally exit. With limited reward feedback, this is particularly difficult to learn. The ability to traverse along $\mathcal{G}_w$ enables longer-horizon planning on top of the waypoints, thanks to which the agents boost the success rate on medium *Door-Key* from $0.56\pm0.02$ to $0.75\pm0.06$.

**Benefits of Learned Waypoints** To highlight the benefit of establishing the waypoints learned by the VAE as nodes for $\mathcal{G}_w$, we compare against results using a $\mathcal{G}_w$ constructed around randomly selected states ($\mathcal{V}_{\mathrm{rand}}$). The edges of the random-node graph are formed in the same way as described in Section 3.3 and its feature extractor is also initialized from $\pi_g$. Although granting knowledge acquired during the unsupervised phase to $\mathcal{V}_{\mathrm{rand}}$ is unfair to $\mathcal{V}_p$, deploying both initialization and traversal while only varying $\mathcal{V}_{\mathrm{rand}}$ and $\mathcal{V}_p$ isolates the effect from the nodes to the best extent. The comparative results (in Table 3, learning curves for *MultiGoal* in Figure 4) suggest $\mathcal{V}_p$ generally outperforms $\mathcal{V}_{\mathrm{rand}}$. *Door-Key* is the only task in which the two matches. However, $\mathcal{V}_{\mathrm{rand}}$ exhibits a large variance, implying that certain sets of random states can be suitable for this task, but using learned waypoints gives strong performance more consistently.

**Initialization with GCP** Initializing the weights of the Worker and Manager feature extractors from $\pi_g$ (learned during the task-agnostic phase) consistently benefits learning. In fact, we observe that models starting from scratch fail on almost all tasks within the maximal number of training iterations, unless coupled with $\mathcal{G}_w$ traversal, which is still inferior to using $\pi_g$-initialization. Particularly, for the small *MultiGoal-Stochastic* environment, there is a high chance that a lava square blocks traversal; therefore, without the environment knowledge from $\pi_g$ transferred by weight initialization, the interference created by the episode-terminating lava prevents the agent from learning the task.

## 6 CONCLUSION

We have shown that world graphs are powerful environment abstractions, which, in particular, are capable of accelerating reinforcement learning. Future works may extend their applications to more challenging RL setups, such as real-world multi-task learning and navigation. It is also interesting to generalize the proposed framework to learn dynamic world graphs for evolving environments, and applying world graphs to multi-agent problems, where agents become part of the world graphs of other agents.

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

## A   RECURRENT VAE WITH DIFFERENTIABLE BINARY LATENT VARIABLES

As illustrated in the main text, the main objective for the recurrent VAE is the following evidence lower bound with derivation:

$$
\begin{aligned}
\log p(a|s) &= \log \int p(a|s,z)dz \\
&= \log \int p(a|s,z)p(z|s)\frac{q(z|a,s)}{q(z|a,s)}dz \\
&= \log \int p(a|s,z)\frac{p(z|s)}{q(z|a,s)}q(z|a,s)dz \\
&\geq \mathbb{E}_{q(z|a,s)}[\log p(a|s,z) - \log \frac{q(z|a,s)}{p(z|s)}] \\
&= \mathbb{E}_{q(z|a,s)}[\log p(a|s,z)] - D_{\mathrm{KL}}(q(z|a,s)||p(z|s))
\end{aligned}
$$

The inference network $q_\psi$ takes in the trajectories of state-action pairs $\tau$ and at each time step approximates the posterior of the corresponding latent variable $z_t$. The prior network $p_\psi$ takes the state $s_t$ at each time step and outputs the state-conditioned prior $p_\psi(s_t)$. We choose Beta as the prior distribution and the Hard Kuma as the approximated posterior to relax the discrete latent variables to continuous surrogates.

The Kuma distribution $\mathrm{Kuma}(\alpha, \beta)$ highly resembles the Beta Distribution in shape but does not come from the exponential family. Similar to Beta, the Kuma distribution also ranges from bimodal (when $\alpha \approx \beta$) to unimodal ($\alpha/\beta \to 0$ or $\alpha/\beta \to \infty$). Also, when $\alpha = 1$ or $\beta = 1$, $\mathrm{Kuma}(\alpha, \beta) = \mathrm{Beta}(\alpha, \beta)$. We observe empirically better performance when we fix $\beta = 1$ for the Kuma approximated posterior. One major advantage of the Kuma distribution is its simple Cumulative Distribution Function (CDF):

$$
F_{\mathrm{Kuma}}(x, \alpha, \beta) = (1 - (1 - x^\alpha))^\beta. \tag{3}
$$

It is therefore amendable to the reparametrization trick (Kingma & Welling, 2013; Rezende et al., 2014; Maddison et al., 2016) by sampling from uniform distribution $u \sim \mathcal{U}(0,1)$:

$$
z = F_{\mathrm{Kuma}}^{-1}(u; \alpha, \beta) \sim \mathrm{Kuma}(\alpha, \beta). \tag{4}
$$

Lastly, the KL-divergence between the Kuma and Beta distributions can be approximated in closed form (Nalisnick & Smyth, 2016):

$$
\begin{aligned}
D_{\mathrm{KL}}(\mathrm{Kuma}(a,b)|\mathrm{Beta}(\alpha,\beta)) &= \frac{a - \alpha}{a}\left(-\gamma - \Psi(b) - \frac{1}{b}\right) \\
&\quad + \log(ab) + \log\mathrm{Beta}(\alpha,\beta) - \frac{b-1}{b} + (\beta - 1)b\sum_{m=1}^{\infty}\frac{1}{m+ab}\mathrm{Beta}\left(\frac{m}{a}, b\right),
\end{aligned} \tag{5}
$$

where $\Psi$ is the Digamma function, $\gamma$ the Euler constant, and the approximation uses the first few terms of the Taylor series expansion. We take the first 5 terms here.

Next, we make the Kuma distribution "hard" by following the steps in Bastings et al. (2019). First stretch the support to $(r = 0 - \epsilon_1, l = 1 + \epsilon_2)$, $\epsilon_1, \epsilon_2 > 0$, and the resulting CDF distribution takes the form:

$$
F_S(z) = F_{\mathrm{Kuma}}\left(\frac{z - l}{r - l}; \alpha, \beta\right). \tag{6}
$$

Then, the non-eligible probabilities for 0's and 1's are attained by rectifying all samples below 0 to 0 and above 1 to 1, and other value as it is, that is

$$
P(z = 0) = F_{\mathrm{Kuma}}\left(\frac{-l}{r - l}; \alpha, \beta\right), \quad P(z = 1) = 1 - F_{\mathrm{Kuma}}\left(\frac{1 - l}{r - l}; \alpha, \beta\right). \tag{7}
$$

Lastly, we impose two additional regularization terms $\mathcal{L}_l$ and $\mathcal{L}_\mathcal{T}$ on the approximated posteriors. As described in the main text, $\mathcal{L}_l$ prevents the model from selecting all states to reconstruct $\{a_t\}_0^{T-1}$

by restraining the expected $L_0$ norm of $\boldsymbol{z} = (z_1 \cdots z_{T-1})$ to approximately be at a targeted value $\mu_0$ (Louizos et al., 2017; Bastings et al., 2019). In other words, this objective adds the constraint that there should be $\mu_0$ of activated $z_t = 1$ given a sequence of length $T$. The other term $\mathcal{L}_{\mathcal{T}}$ encourages temporally isolated activation of $z_t$, meaning the number of transition between 0 and 1 among $z_t$'s should roughly be $2\mu_0$. Note that both expectations in Equation 2 have closed forms for HardKuma.

$$\mathcal{L}_0 = \left\| \mathbb{E}_{q(\boldsymbol{z}|\boldsymbol{s},\boldsymbol{a})} \left[ \|\boldsymbol{z}\|_0 \right] - \mu_0 \right\|^2, \text{ where} \tag{8}$$

$$\mathbb{E}_{q(\boldsymbol{z}|\boldsymbol{s},\boldsymbol{a})} \left[ \|\boldsymbol{z}\|_0 \right] = \sum_{t=1}^{T} \mathbb{E}_{q(z_t|\boldsymbol{s},\boldsymbol{a})} \left[ \mathbb{1}_{z_t \neq 0} \right]$$

$$= \sum_{t=1}^{T} 1 - p\left(z_t = 0\right) = \sum_{t=1}^{T} 1 - F_{\text{Kuma}}\left(\frac{-l}{r-l}; \alpha_t, \beta_t\right), \tag{9}$$

$$\mathcal{L}_T = \left\| \mathbb{E}_{q(\boldsymbol{z}|\boldsymbol{s},\boldsymbol{a})} \sum_{t=1}^{T-1} \mathbb{1}_{z_t \neq z_{t+1}} - 2\mu_0 \right\|^2, \text{ where} \tag{10}$$

$$\mathbb{E}_{q(\boldsymbol{z}|\boldsymbol{s},\boldsymbol{a})} \left[ \sum_{t=1}^{T-1} \mathbb{1}_{z_t \neq z_{t+1}} \right] = \sum_{t=1}^{T-1} p\left(z_t = 0\right)\left(1 - p\left(z_{t+1} = 0\right)\right) + \left(1 - p\left(z_t = 0\right)\right)p\left(z_{t+1} = 0\right). \tag{11}$$

**Lagrangian Relaxation.** The overall optimization objective consists of action sequence reconstruction, KL-divergence between the posterior and prior, $\mathcal{L}_0$ and $\mathcal{L}_T$ (Equation 12). We tune the objective weights $\lambda_i$ using Lagrangian relaxation (Higgins et al., 2017; Bastings et al., 2019; Bertsekas, 1999), treating $\lambda_i$'s as learnable parameters and performing alternative optimization between $\lambda_i$'s and the model parameters. We observe that as long as their initialization is within a reasonable range, $\lambda_i$'s converge to a local optimum:

$$\max_{\{\lambda_{1,2,3}\}} \min_{\{\theta,\phi,\psi\}} -\mathbb{E}_{q_\psi(\boldsymbol{z}|\boldsymbol{a},\boldsymbol{s})} \left[ \log p_\theta(\boldsymbol{a}|\boldsymbol{s},\boldsymbol{z}) \right] + \lambda_1 D_{\text{KL}}\left(q_\phi(\boldsymbol{z}|\boldsymbol{a},\boldsymbol{s}) | p_\psi(\boldsymbol{z}|\boldsymbol{s})\right) + \lambda_2 \mathcal{L}_0 + \lambda_3 \mathcal{L}_T. \tag{12}$$

We observe this approach to produce efficient and stable mini-batch training.

## B GOAL-CONDITIONED POLICY INITIALIZATION FOR HRL

Optimizing composite neural networks like HRL (Co-Reyes et al., 2018) is sensitive to weight initialization (Mishkin & Matas, 2015; Le et al., 2015), due to its complexity and lack of clear supervision at various levels. Therefore, taking inspiration from prevailing pre-training procedures in computer vision (Russakovsky et al., 2015; Donahue et al., 2014) and NLP (Devlin et al., 2018; Radford et al., 2019), we take advantage of the weights learned by $\pi_g$ during world graph discovery when initializing the Worker and Manager policies for downstream HRL, as $\pi_g$ has already implicitly embodied much environment dynamics information.

More specifically, we extract the weights of the feature extractor, i.e. the state encoder, and use them as the initial weights for the state encoders of the HRL policies. Our empirical results demonstrate that such weight initialization consistently improves performance and validates the value of skill/knowledge transfer from GCP (Taylor & Stone, 2009; Barreto et al., 2017).

## C ADDITIONAL IMPLEMENTATION DETAILS

Model code folder including all architecture details is shared in comment.

### C.1 HYPERPARAMETERS FOR VAE TRAINING

Our models are optimized with Adam (Kingma & Ba, 2014) using mini-batches of size 128, thus spawning 128 asynchronous agents to explore. We use an initial learning rate of 0.0001, with $\epsilon = 0.001, \beta_1 = 0.9, \beta_2 = 0.999$; gradients are clipped to 40 for inference and generation nets. For HardKuma, we set $l = -0.1$ and $r = 1.1$. The maximum sequence length for BiLSTM is 25. The

total number of training iterations is 3600 and model usually converges around 2400 iterations. We train the prior, inference, and generation networks end-to-end.

We initialize $\lambda_i$'s (see **Lagrangian Relaxation**) to be $\lambda_1 = 0.01$ (KL-divergence),$\lambda_2 = 0.06$ ($\mathcal{L}_0$), $\lambda_3 = 0.02$ ($\mathcal{L}_T$). After each update of the latent model, we update $\lambda_i$'s, whose initial learning rate is 0.0005, by maximizing the original objective in a similar way as using Lagrangian Multiplier. At the end of optimization, $\lambda_i$'s converge to locally optimal values. For example, with the medium maze, $\lambda_1 = 0.067$ for the KL-term, $\lambda_2 = 0.070$ for the $\mathcal{L}_0$ and $\lambda_3 = 0.051$ for the $\mathcal{L}_T$ term. The total number of waypoints $|\mathcal{V}_p|$ is set to be 20% of the size of the full state space.

## C.2   TRAINING HRL MODELS

The procedure of the Manager and the Worker in sending/receiving orders using either traversal paths among $\mathcal{V}_p$ from replay buffer for deterministic environments or with $\pi_g$ for stochastic ones follows:

1. The Manager gives a wide-narrow subgoal pair $(g_w, g_n)$.
2. The agent takes action based on the Worker policy $\pi^\omega$ conditioned on $(g_w, g_n)$ and reaches a new state $s'$. If $s' \in \mathcal{V}_p$, $g_w$ has not yet met, and there exists a valid path basing on the edge paths from the world graph $s' \to g_w$, agent then either follows replay actions or $\pi_g$ to reach $g_w$. If $\pi_g$ still does not reach desired destination in a certain steps, then stop the agent wherever it stands; also $\pi_g$ can be finetuned here.
3. The Worker receives positive reward for reaching $g_w$ for the first time.
4. If agent reaches $g_n$, the Worker also receives positive rewards and terminates this horizon.
5. The Worker receives negative for every action taken except for during traversal; the Manager receives negative reward for every action taken including traversal.
6. When either $g_n$ is reached or the maximum time step for this horizon is met, the Manager renews its subgoal pair.

The training of the Worker policy $\pi^\omega$ follows the same A2C algorithm as $\pi_g$.

The training of the Manager policy $\pi^m$ also follows a similar procedure but as it operates at a lower temporal resolution, its value function regresses against the $t_m$-step discounted reward where $t_m$ covers all actions and rewards generated from the Worker.

When using the Wide-then-Narrow instruction, the policy gradient for the Manager policy $\pi_m$ becomes:

$$\mathbb{E}_{(s_t, a_t) \sim \pi, p, p_0} \left[ A_{m,t} \nabla \log \left( \pi^\omega \left( g_{w,t} | s_t \right) \pi^n \left( g_{n,t} | s_t, g_{w,t}, s_{w,t} \right) \right) \right] + \nabla \left[ \mathcal{H} \left( \pi^\omega \right) + \mathcal{H} \left( \pi^n (\cdot | g_{w,t}) \right) \right],$$

where $A_{m,t}$ is the Manager's advantage at time $t$. Also, for Manager, as the size of the action space scales linearly with $|\mathcal{S}|$, the exact entropy for the $\pi^m$ can easily become intractable. Essentially there are $O\left( |\mathcal{V}| \times (N^2) \right)$ possible actions. To calculate the entropy exactly, all of them has to be summed, making it easily computationally intractable:

$$\mathcal{H} = \sum_{w \in \mathcal{V}} \sum_{w_n \in s_w} \pi^n(w_n | s_w, s_t) \pi^\omega(w | s_t) \log \nabla \pi^n(w_n | s_w, s_t) \pi^\omega(w | s_t).$$

Thus in practice we resort to an effective alternative $\mathcal{H} \left( \pi^\omega \right) + \mathcal{H} \left( \pi^n (\cdot | g_{w,t}) \right)$.

Psuedo-code for Manager training is in Algorithm 2.

## C.3   HYPERPARAMETERS FOR HRL

For training the HRL policies, we inherit most hyperparameters from those used when training $\pi_g$, as the Manager and the Worker both share similar architectures with $\pi_g$. The hyperparameters used when training $\pi_g$ follow those from Shang et al. (2019). Because the tasks used in HRL experiments are more difficult than the generic goal-reaching task, we set the maximal number of training iterations to 100K abd training is stopped early if model performance reaches a plateau. The rollout steps for each iteration is 60. Hyperparameters specific to HRL are the horizon $c = 20$ and the size of the Manager's local attention range (that is, the neighborhood around $g^w$ within which $g^n$ is selected), which are $N = 5$ for small and medium mazes, and $N = 7$ for the large maze.

---

**Algorithm 2:** Training of $\pi^m$ for HRL models

---

Initialize network parameters $\theta$ for $\pi^m$, here $\pi^{m,t}$ refers to the policy at time rollout time step $t$;
Given a map of $\mathcal{V}$, $s_{\mathcal{V}}$;
**for** iter $= 0, 1, 2, \cdots$ **do**
   Clear gradients $d\theta \leftarrow 0$;
   Reset the set of time steps where $\pi^{m,t}$ omits a new subgoal $S_m = \{\}$ and $t_m = 0$.;
   **while** $t <= t_{\max}$ *or episode not terminated* **do**
      Simulate under current policy $\pi^{m,t-1}, \pi^{w,t-1}$;
      **if** *the Worker has met the previous subgoal or exceeded the horizon c* **then**
         Sample a new subgoal $g_{m,t}$ from $\pi^{m,t}$;
         $z_{m,t} = f_{\text{LSTM}}(\text{CNN}(s_{m,t}, s_{\mathcal{V}}), h_{m,t_m})$, $V_{m,t} = f_v(z_{m,t})$, $\pi_t = f_p(z_{m,t})$ ;
      **end**
      $S_m = S_m \cup \{t_m\}$ and $t_m = t$;
   **end**
   $R = \begin{cases} 0, & \text{if terminal} \\ V_{t_{\max}+1}, & \text{otherwise} \end{cases}$;
   **for** $t = t_{\max}, \cdots 1$ **do**
      $R \leftarrow r_t + \gamma R$;
      **if** $t \in S_m$ **then**
         $A_{m,t} \leftarrow R - V_{m,t}$;
         Accumulate gradients from value loss: $d\theta \leftarrow d\theta + \lambda \frac{\partial A_{m,t}^2}{\partial \theta}$;
         Accumulate policy gradients with entropy regularization:
            $d\theta \leftarrow d\theta + \nabla \log \pi_{m,t}(g_{m,t}) A_{m,t} + \beta \nabla H(\pi_{m,t})$;
      **end**
   **end**
**end**

---

## D    2D GRID WORLD VISUALIZATIONS

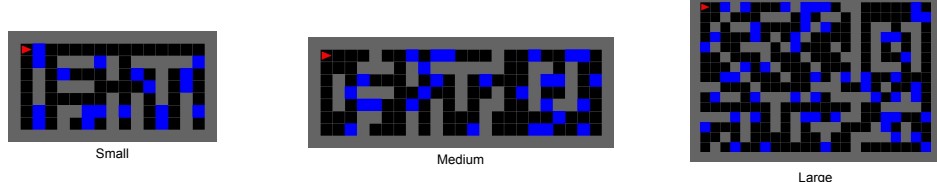

Figure 5: Visualization of the 2D grid environments in our experiments, along with the learned waypoints in blue.

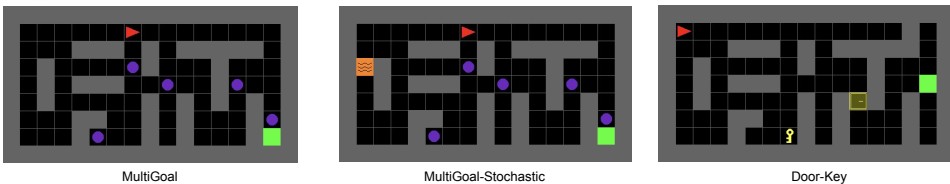

Figure 6: Visualization of tasks in our experiments.

