# OpenReview forum: "Learning World Graph Decompositions To Accelerate Reinforcement Learning"
_ICLR.cc/2020/Conference — Reject_

### Official Review · AnonReviewer2 · 2019-10-22
**Official Blind Review #2**

**Rating:** 6

**Review:**

This paper proposes a novel approach to hierarchical reinforcement learning approach by first learning a graph decomposition of the state space through a recurrent VAE and then use the learned graph to efficiently explore the environment. The algorithm is separated into 2 stages where in the first stage random walk and goal conditioned policy is used to explore the environment and simultaneous use a recurrent binary VAE to compress the trajectory. The inference network is given the observation and action and the reconstruction is to, given the hidden state or hidden state+observation, reconstruct the action taken. The approximate posterior takes on the form of a hard Kumaraswamy distribution which can differentiably approximate a binary variable; when the approximate posterior is 0, the decoder must reconstruct the action using the hidden state alone. The nodes of the world graph are roughly states that are used to reconstruct the trajectories in the environment. After the graph is constructed, the agent can use a combination of high-level policy and classical planning to solve tasks with sparse reward.

Personally, I quite like the idea of decomposing the world into important states -- it is closely related to the concept of empowerment [1] which the authors might want to take a further look into. I believe extracting meaningful abstraction from the environment will be a key component for general purpose RL agent. One concept I really like in the paper is using the reconstruction error as the reward for the RL agent, which has some flavors of adversarial representation learning. Further, I also really like the idea of doing structured exploration in the world graph and I believe doing so can help efficiently solve difficult tasks.

However, I cannot recommend accepting this paper in its current draft as there might be potential major technical flaw and I also have worries about the generality of the algorithm. My main concerns are the following:
    1. The ELBO given in the paper is wrong -- the KL divergence should be negative. I want to give the paper the benefit of doubts since this could be just a typo and some (very rare) researchers use a reverse convention; however, this sign is wrong everywhere in the paper including the appendix yet the KL between Kuma and beta distributions uses the regular convention. I tried to check the source code provided by the authors but the code only contains architecture but not the training objective, training loops or environments. As such, I have to assume that the ELBO was wrongfully implemented, unless the author can provide the full source code, or, if the ELBO is indeed incorrectly implemented, rerun the experiments with the correct implementation.

    2. The proposed method for learning the graph does not have to be a VAE at all. The appendix shows that the paper uses a 0.01 coefficient on the KL, which is an extremely small value for VAE (in fact, most VAE’s have beta larger than 1 for disentanglement). Thus, I suspect the KL term is not actually doing anything; instead, the main reason why the model worked might be due to the sparsity constraints L_0 and L_T. In other words, the model is simply behaving like a sequence autoencoder with some sort of hard attention mechanism on the hidden code, which might explain why the model still worked well even with the wrong ELBO. To clarify, I think this is a perfectly acceptable approach for learning the graph and it would still be very novel, but the manuscript should be revised accordingly to reflect this. If the (fixed) VAE is important, then this comparison (0 KL regularization) would be a nice ablation regardless.

    3. Algorithm 1 requires navigating the agent to the key points from \mathcal{V}_p. This assumption is quite strong. When the transition dynamic is deterministic and fully reversible like the ones considered in the paper, using the reverse of replay buffer can indeed take the agent back to s_p, but in settings where the transitions are stochastic or the transitions are non-linear or non-reversible, how should the algorithm be used?

    4. It is not clear how \mathcal{V}_p are maintained. If multiple new nodes are added every iteration, wouldn't there be more than necessary nodes in \mathcal{V}_p? It seems to me some pruning criteria were used unless the model converged within small number of iterations? Are the older ones are discarded in favor of newer ones?

    5. How are the actions sequences “normalized”?

    6. In what way are the Door-Key environment stochastic? It seems like the other environments also have randomness, so is the only difference the lava pool?

I believe the propose method is sound, so if the revision can address either 1 or 2, I am willing to raise my score to weakly accept. If the revision in addition addresses 3, 4, 5, 6 in a reasonable manner, I am willing to raise my score to accept.

=======================================================================
Minor comments that did not affect my decision:
    - I think mentioning the names of the environment in the abstract might be uninformative since the readers do not know what they are a priori.

Reference:
[1] Empowerment -- An Introduction, Salge et al. 2014


**Experience Assessment:**

I have published one or two papers in this area.

**Review Assessment: Checking Correctness Of Derivations And Theory:**

I carefully checked the derivations and theory.

**Review Assessment: Checking Correctness Of Experiments:**

I carefully checked the experiments.

**Review Assessment: Thoroughness In Paper Reading:**

I read the paper thoroughly.

---

> ### Author Response · Authors · 2019-11-13
> **Thank you very much for your feedback!**
>
> We thank you for the reviews and feedback!
>
> 1. ELBO formulation: Thank you very much for spotting the typo. The objective used for training uses the correct ELBO. We have corrected the main text and appendix and also uploaded the relevant training code (in PriorLoss.py, we are indeed minimizing the KL divergence).
>
> 2. KL term: The VAE model presents three essential advantages: (1) it reflects the intrinsic stochasticity that given a trajectory, there can be multiple combinations of intermediate states capable of conveying sufficient information for action recovery. (2) our prior reflects the empirical average selection probabilities for each state, meaning it encodes the average activations yielded by the inference network. Regularizing the approximated posterior with the prior (main text 3.1) encourages each trajectory to choose the combination consisting of frequently activated states to activate. (3) We also leverage the prior mean in selecting the waypoints (last line Algorithm 1).
>
> Because the regularization imposed by the KL term is fairly significant and in our case the prior is learned upon how often a state is selected,  an overly aggressive KL risks creating a constrained feedback cycle in the learning dynamics of these network components, which would cause the VAE to prematurely converge. For the same reason, many VAE applications--especially when not involving sampling from the prior--sets the KL term coefficient to a small value, e.g. the STOA poseVAE [1] has it as low as 0.0005.
>
> 3. The sole purpose of navigating to waypoint states before performing exploration is to allow for the starting positions of exploration rollouts to expand as the agent discovers more of its environment. There are numerous strategies to achieve this when stochasticity limits the usefulness of memorized action sequences. One option is to use the goal-conditioned policy (GCP) to navigate to the target waypoint state and set wherever the GCP ends up as the starting point for exploration, since the precision of starting points is unimportant. All that we wish to ensure is that the range of starting points expands as training progresses.
>
> We have revised the main text (3.2) to make it clear that our choice of replaying action sequences relies on the deterministic dynamics of our testbed environment and that other choices may be appropriate in different settings. We also clarify that the details of such a choice are likely not crucial provided that they achieve the goal of expanding the range of exploration starting points.
>
> 4. $\mathcal{V}_p$, the set of waypoint states, is updated after every few iterations based on a ranking using the prior mean (last line in Algorithm 1) with the top 20% selected into $\mathcal{V}_p$ (Appendix C.1 specifies this hyperparameter). Waypoint states are not necessarily carried over from one iteration to the next. That is, we re-compute the set of waypoint states based on the prior network updates. We have revised the main text (3.1) to make this clearer.
>
> 5. We correct the main text and removed “normalized” as our normalization (mean subtraction and standard deviation division) does not affect the actual execution of the algorithm.
>
> 6. Different minigrid tasks are described in Table 1; particularly, for Door-Key, the location of the door, the key, the exit point and an additional wall block are generated at random. On the small maze alone, there are approximately 70K variations.
>
> [1] Walker et al.  The Pose Knows: Video Forecasting by Generating Pose Futures.

---

> > ### Comment · AnonReviewer2 · 2019-11-14
> > **Thank you for the clarification.**
> >
> > Thank you for the clarification, and the effort to update the paper/provide the code.
> >
> > 1. I will trust that authors were using the correct ELBO, and raise my score to weakly accept as promised; however, I want to point out that the new code only added a few new pieces and many imports and libraries used by the scripts are not included.
> >
> > 2. Thank you for your explanation regarding the usage of VAE, but I am still not convinced by the usage of small coefficient. It is common to use KL annealing in sequence generation tasks [1] but ultimately the weight is increased to 1. One of the first usage of VAE in sequential modeling [2] also uses lambda = 1 for the KL divergence. I am not an expert in pose estimation so I do not wish to comment on the paper authors refer to  (which also uses a form of KL annealing) but as far as I can tell the application is not close to the task this paper tries to accomplish. I would be much more convinced if authors can provide some more relevant references and/or provide an ablation study on the choice of lambda in the final version. Since the method aims to be a generic HRL algorithm, the bar for picking hyperparameters should be higher than application specific papers.
> >
> > 3. While your proposal seems reasonable, no experiments (even if the performance is sub-optimal) are provided to back up the claim. In particular, you propose to use GCP to accomplish the same purpose but when GCP is not properly trained (e.g. at the beginning of training), the GCP will not be very good, and this may affect the quality of the final policy adversely, if the policy even learns at all. I find it unconstructive to extrapolate what the algorithm would do in completely different domains. As it stands, I don’t believe this concern has been addressed.
> >
> > 4. Thank you for clarifying this. It seems like 20% is extremely high and may be an obstacle for generalizing this approach to other domains. Can you provide ablation on the proportion? Another problem is that if the policy cannot sufficiently cover the entire state space during the cycle, the $\mathcal{V}_p$ won’t be good enough. I have minor concerns about whether this would scale. However, this may be addressed with engineering and some sort of curiosity or other form of exploration bonus.
> >
> > 5. Fair enough.
> >
> > 6. I understand how door-key environment is set up but it seems all of the environments that are not multigoal have some components of stochasticity, hence my confusion. Further, are the environments also stochastic during graph learning? Or are they only stochastic during the hierarchical policy learning?
> >
> > Additional questions: The assumption that the agent can recognize encountering a waypoint is also interesting. How is this implemented? Does this involve searching over all the waypoints and measure the L2 distance? Will this have scaling bottleneck? Can this be solved with some parametric model?
> >
> > A part of my concerns has been addressed, but some remain. I look forward to hearing your response to the rest.
> >
> > [1] Generating Sentences from a Continuous Space. Bowman et al.
> > [2] A Recurrent Latent Variable Model for Sequential Data. Chung et al.

---

> > > ### Author Response · Authors · 2019-11-15
> > > **Thank you again for your detailed feedback! [part 1]**
> > >
> > > 1. Thank you! The shared training code reflects the essence of our algorithm. Many of the imports and helper files have not been formatted to be informative, but are not essential for understanding how the algorithms work. We will share the full cleaned code upon acceptance.
> > >
> > > 2. In fact, it is not practical to compare VAEs used in [1] and [2] to ours head to head because the application, data domain, latent space type, prior, approximated posterior are all different.
> > >
> > > More importantly, many recent efforts have devoted to studying the effect of KL term. In particular, [3] contributes a nice ablation study over this topic and recommends a rule of thumb. In general, when choosing the KL weight:
> > > - If one’s goal is to perform log-likelihood maximization and log-likelihood is the measurement of model performance--which is indeed the case in [1] and [2], then KL should be set to 1. Otherwise it violates the consistency between training objective and evaluation objective
> > > - However, if one’s ultimate goal is not log-likelihood, that is, if the model’s generative property is not the primary goal, the KL-term is often regarded as a regularization term and can be set as application-appropriate. For instance, in the case where one wants to prevent an overly strong decoder that can ignore latent code [3] or if the regularization from KL term ([6] and in our case) is so strong that it can cause posterior collapsing, the KL term is set to smaller than 1. On the other hand, if one desires disentangling property such as in beta-VAE, then the KL term is set to bigger than 1 [4].
> > >
> > > Lastly, we’d like to draw R4’s attention to the last paragraph in Appendix A. In selecting hyperparameters, we in fact leverage a neat technique, Lagrangian Relaxation, which allows the weights between different losses to adaptively balance among one another. In other words, as long as our initial hyperparameter settings for the loss coefficients are within a reasonable range, the coefficients automatically converge to a local optima without manual annealing. For example, for the medium maze, our initial KL term weight is set to be 0.01 and it converges to 0.067 at the end of training. This optimization technique also has been used by previous work [5], which should serve as the most relevant basis for comparison to our approach given their application of the HardKuma distribution and regularization of sequence statistics.
> > >
> > > Nevertheless, we are happy to more exhaustively demonstrate the impact of initial weights given to the KL term for the final version.
> > >
> > > [1] Generating Sentences from a Continuous Space. Bowman et al.
> > > [2] A Recurrent Latent Variable Model for Sequential Data. Chung et al.
> > > [3] Fixing a Broken ELBO, Alemi et al.
> > > [4] β-VAE: Learning Basic Visual Concepts with a Constrained Variational Framework, Higgins et al, 2017.
> > > [5] Interpretable Neural Predictions with Differentiable Binary Variables,
> > > [6] The Pose Knows: Video Forecasting by Generating Pose Futures, Walker et al.
> > >
> > > 3. We are happy to more rigorously evaluate modifications of our approach for learning world graphs that may better accommodate stochasticity, as part of the final version. In particular, we plan to run experiments where, during world graph learning, we ignore action replays for reaching waypoint states and instead use a strategy such as that proposed above (i.e. using the GCP). We expect this to work based on the following intuition:
> > > - Our GCP is trained by using a form of exploration bonus, i.e., rewarding diversity in the states that are reached, and so motivates agents to expand the set of visited states. This automatically expands the set of potential goals. Exploration-based reward shaping techniques include, e.g., intrinsic motivation, curiosity, etc, which have been successfully used to learn to play Atari games (e.g., Montezuma’s Revenge) [1], continuous control [2], and others [3]. In all these applications, it has been shown that rewarding diversity biases agents to discover e.g., states with high rewards faster.
> > > - Another intuition is that such GCP agents have visited good waypoint states that could be identified by our world graph algorithm. Consider an agent that can solve Montezuma’s Revenge thanks to having learned a good exploration policy. Every 10th frame selected from a successful state sequence that ends in winning the game would constitute a good (partial) world graph for the (single) task of Montezuma’s Revenge. This suggests using GCPs with exploration bonuses would yield world graphs that are useful for multiple tasks as well. Hence, we expect our iterative refinement approach to learn the world graphs would be effective in other applications as well.
> > >
> > > [1] Unifying Count-Based Exploration and Intrinsic Motivation, Marc G. Bellemare et al,
> > > [2] Large-Scale Study of Curiosity-Driven Learning, Buda et al 2018,
> > > [3] Exploration by Random Network Distillation. Burda et al 2018

---

> > > ### Author Response · Authors · 2019-11-15
> > > **Thank you again for your detailed feedback! [part 2]**
> > >
> > > 4. We are happy to add more comparisons with various waypoint selection rates and neighborhood sizes. To clarify, ‘neighborhood size’ refers to the size of the neighborhood around a waypoint state (wide goal) within which the WN Manager can propose the narrow goal. Intuitively, if the proportion of waypoint states and/or the size of the neighborhood is too small, it may create “blind spots” in the state space such that there are states that the WN Manager cannot select as narrow goals.
> > > We chose the neighborhood size and waypoint selection rate based on this intuition and did not tune them during our experiments. Hence, we expect our algorithm to be fairly robust to the choice of waypoint state density. For lower selection rates (< 20%), one can increase the area considered for “narrow goals” by the Manager (that is, the neighborhood size) to compensate so as to ensure a sufficient coverage of the entire environment. Again, we will experiment with different combinations of of waypoint selection rates and neighborhood sizes and present the results as part of the final version.
> > > We agree that scaling up to other domains may require further refinements of our approach, but we see this as very compelling future work.
> > >
> > > 6. The stochasticity comes from specific tasks. Hence, in the world graph phase, there is no stochasticity in our case. It is only present in the HRL phase (for each specific task that we used). Note that all tasks feature certain type of stochasticity.
> > >
> > > Answers to additional questions: For discrete state spaces, this is simply a constant-time dictionary lookup. For continuous state spaces, one can use L2-distance epsilon-balls around waypoints. As long as the set of waypoint-neighborhoods covers the state space, and we assume a fixed fraction of neighborhoods / state-space volume, this scales linearly with the volume of the state space; hence, it would likely scale well.

---

### Official Review · AnonReviewer3 · 2019-10-23
**Official Blind Review #3**

**Rating:** 3

**Review:**

Summary: This paper proposes an approach to identifying important waypoint states in RL domains in an unsupervised fashion, and then for using these states within a hierarchical RL approach.  Specifically, the authors propose to use a binary latent variable VAE to identify waypoint states, then an HRL algorithm uses these waypoints as intermediate goals to better decompose large RL domains. The authors show that on several grid world tasks, the resulting policies substantially outperform baseline approaches.

Comments: I have mixed opinions on this paper, though ultimately feel that it is below the bar for acceptance.  There are a lot of aspects to the proposed approach, and overall I'm left with the impression that there are some interesting and worthwhile aspects to the proposed method, but ultimately it is hard to disentangle precisely what is the effect of each contribution.

For example, let's consider the waypoint discovery method.  The basic approach is to use a binary latent variable VAE, using a recently proposed Hard Kumaraswamy distribution to model the latent state.  This seems like a reasonable approach, but there's no real analysis of the actual waypoints that are discovered in the target domains, whether they indeed correspond to intuitively important waypoints in a domain, or whether they are just producing some arbitrary segmentation of the proposed task.

The other elements of the paper have similar issues for me.  The whole HRL process, using these waypoint states as intermediate goals, seems reasonable, but it's hard to disentangle the performance of this particular approach versus the performance of any approach that would use (any) intermediate states as goals within an HRL approach.  And the impression I'm left with, given the level of detail included I the paper, is that I would have no idea how to apply or extend this process to any other RL domains.

I looked at the provided code hoping it would help to clarify some of the implementation details, but the code is not at all a complete collection of routines that could re-create the experiments.  Rather, the code just includes a few of the model architectures, which aren't really the important aspects of this work.

Thus, I'm overall left with the impression that it's quite difficult to assess the contribution of this approach, and determine precisely which of the different proposed aspects is really contributing most to the improved performance.  I know there is some ablative analysis in the paper comparing the pi_g-init and G_w-traversal independently and together, but I'm more questioning the basic question of what each portion of the network is really learning.

I'd be curious if the authors are able to clarify any of these points during their rebuttal.

**Experience Assessment:**

I have published in this field for several years.

**Review Assessment: Checking Correctness Of Derivations And Theory:**

I assessed the sensibility of the derivations and theory.

**Review Assessment: Checking Correctness Of Experiments:**

I assessed the sensibility of the experiments.

**Review Assessment: Thoroughness In Paper Reading:**

I read the paper thoroughly.

---

> ### Author Response · Authors · 2019-11-13
> **Thank you for your feedback!**
>
> We thank you for your feedback. Please see the general comment that clarifies our contributions and how our ablation and comparative analysis shows the impact of the various components of our approach. We have included the core training code to clarify how training is implemented.
>
>
> “there's no real analysis of the actual waypoints that are discovered in the target domains, whether they indeed correspond to intuitively important waypoints in a domain, or whether they are just producing some arbitrary segmentation of the proposed task.”
> -In fact, we show samples of learned waypoints in Appendix D. These show that the learned waypoints represent intuitive decompositions of the mazes and important states, e.g., hallway segments, junctions.
>
>
> “but it's hard to disentangle the performance of this particular approach versus the performance of any approach that would use (any) intermediate states as goals within an HRL approach.“
> -To validate the use of learned waypoints in a world graph, we have 2 comparisons. These show that using learned waypoints is significantly better than other waypoints selection methods.
>
> First, we compare using world graphs with **randomly** selected waypoints on learning downstream tasks. For example, Table 3 shows that when our HRL approach uses graphs with waypoint states selected by the VAE performance is better and more consistent than when using graphs with randomly selected states as its nodes.
>
> Secondly, we compare learned waypoints with a Feudal Network that allows the Manager network to **select any state as a goal** (Table 2). On our tasks, this network only barely does better than a vanilla A2C baseline.
>
>
> “And the impression I'm left with, given the level of detail included in the paper, is that I would have no idea how to apply or extend this process to any other RL domains.”
> -We have updated Appendix C, detailing step-by-step procedures of our HRL algorithm for the task-specific phase. We also kindly encourage the reviewer to inspect the shared code for the implementation of our approach. Although our results stand on their own as validation of our approach, we believe that extending our approach to other domains is a fruitful direction for future work.
>
>
> "Thus, I'm overall left with the impression that it's quite difficult to assess the contribution of this approach, and determine precisely which of the different proposed aspects is really contributing most to the improved performance.  I know there is some ablative analysis in the paper comparing the pi_g-init and G_w-traversal independently and together, but I'm more questioning the basic question of what each portion of the network is really learning."
> -Please see the general comment for an overview of our work. Our core contributions are a framework to learn world graphs (Section 3) and how to use them effectively for structured exploration (Section 4). We analyzed the impact of each aspect of our framework in our ablation studies.
>
> We understand the complete framework (graph learning + HRL) has a number of novel methodological features, which could be hard to disentangle. We found empirically that each part of our approach is essential to “make it work”, i.e., learning useful world graphs and effectively utilizing them to accelerate HRL. Our ablation studies thoroughly show how each component impacts performance.
>
> Of course, each component in our work can be further developed. We very much hope to see our work stimulating future research endeavors and provide a testbed and baseline benchmark.
>
> We will clarify the writing to show how the comparative analysis inspects the various parts of our framework.

---

### Official Review · AnonReviewer4 · 2019-10-29
**Official Blind Review #4**

**Rating:** 3

**Review:**

The paper proposes an interesting method to construct a world graph of helpful exploration nodes to provide “structured exploration”. This graph is used in an HRL structure based on the feudal net structure. While the method is very interesting the proposed method is designed to help learn good policies via a better exploration structure. This is a very important problem but I find that the environments this method is tested on in the paper can be easily solved using normal RL methods. It would be very important to evaluate the methods progress on more interesting problems with complex temporal structure. Potentially some of the tasks from the HIRO paper or better yet an assembly tasks or version of the clevr objects environment where multiple items need to be rearranged into a goal. One of the more advanced tasks from the Hierarchical Actor-Critic paper would also be a good option. It is also important to include more analysis of the amount of data needed to train the VAE and create the graph. This amount should be included in the evaluation results for the method.

More detailed comments:
-	The last paragraph of the introduction that begins to explain the method is a bit confusing. More detail here would be helpful. How frequently do binary latent variables need to be selected for them to become nodes? Similar for adding edges.
-	In Figure 1, there are many terms that have not been defined yet, "pivotal state", "world graph traversal"... It would help in understanding the figure if these were explained beforehand. The figure text is also very small.
-	This work seems very similar to Search on the replay buffer (Eysenbach et al 2019). That work created a graph over the data in the replay buffer based on the Q value of different states. These act as waypoints in planning. Could this method not be used to also construct a more sparse waypoint graph to use such as what is described in this work?
-	It is said that the primary goal of the graph is to accelerate downstream tasks. Yet, the graph is constructed with states that are most critical in recovering action sequences.  Is there some guarantee that this selection criterion will help downstream tasks?
-	The method also seems to have a similarity to the GoExplore paper that keeps around an exploration frontier, that is similar to the world graph, of states as it is making progress on the task. This paper should be discussed in more detail in the related work.
-	The VAE is trained over data that is collected from the policy during exploration. Is there an issue with collecting data that will extrapolate to explore areas of the state space that are outside of the data collected for training the VAE.
-	More detail should be included in the use of \mu_0. As it is written now it is difficult for the reader to understand how the method works without some of the additional information in the appendix.
-	The method to collect enough data to learn and represent a graph the covers the state space well. How well does this method work? Essentially this method is making progress on the exploration problem. Is there some analysis on how well this method is at collecting enough data to use on downstream tasks?
-	The method uses curiosity to help explore the state space by using the reconstruction error from the VAE as an intrinsic reward. Is a version of this intrinsic reward use for the baseline A2C method in the paper?
-	It is said that the world graph helps accelerate learning via structured exploration. However, there is there a significant amount of compute and environment interaction to compute the world graph? This should be taken into consideration when performing any comparisons.
-	Can the graph be updated during the policy training phase? Also, in the first phase where data is collected to fit the VAE, can this data be used to train an off-policy method? It seems like this data would work very well for training a policy.
-	Table 3 does not seem to be referenced in the paper. They could also use some additional explination as to what the values represent that are being presented.


**Experience Assessment:**

I have published in this field for several years.

**Review Assessment: Checking Correctness Of Derivations And Theory:**

I assessed the sensibility of the derivations and theory.

**Review Assessment: Checking Correctness Of Experiments:**

I carefully checked the experiments.

**Review Assessment: Thoroughness In Paper Reading:**

I read the paper thoroughly.

---

> ### Author Response · Authors · 2019-11-13
> **Thank you very much! Rebuttal Part 1**
>
> We thank you for your feedback and for acknowledging the importance of the problem tackled in our work.
>
> We respectfully disagree that our benchmark tasks, especially those with more challenging setups, can be “easily” solved via standard baselines. For instance, Table 2 shows that the baselines A2C and its hierarchical variant Feudal Network only manage to solve MultiGoal and MultiGoal-Stochastic on the small mazes. For those small mazes, the final performance is worse than when using graph learning. The baselines fail to solve the larger mazes within the number of samples. Figure 4 further highlights that incorporating graph learning significantly speeds up the baselines.
>
> We also thank R4 for the environment suggestions. However, we feel that the current suite of evaluation tasks provide sufficient evidence of the merits of our approach. Our used maze environment and tasks validate a key hypothesis: that our learned world graph approach enables RL agents to solve a **diverse suite of tasks** that require understanding of the structure of the environment (e.g., how to navigate in a maze).
>
> Moreover, the grid-world environment enables clear exposition, which is crucial in evaluating the various parts of our framework. Note that other work that introduces new learning approach, e.g. [1], similarly use clean environments to remove confounding factors for analysis.
>
> [1] T. Kipf Compositional Imitation Learning: Explaining and Executing One Task at a Time.

---

> > ### Comment · AnonReviewer4 · 2019-11-13
> > **Thank you for your clarifications**
> >
> > Thank you for your clarifications
> >
> > These comments have helped clear up my understanding of some important details.

---

> ### Author Response · Authors · 2019-11-13
> **Rebuttal Part 2 :)**
>
> -Last paragraph of intro and nodes (i.e. waypoints) updates:
> The description in the intro is meant to convey the high-level idea, with specifics provided in the methods. Method details are in section 3: waypoint identification 3.1 and edge formation 3.3. Particularly, the set of waypoint states, is updated after every few iterations based on a ranking using the prior mean (last line in Algorithm 1) with the top 20% selected. The set of edges are forged after the set of waypoints is finalized.
>
> -Figure1:
> Thank you for pointing this out. Figure1 intends to provide an intuitive example and guide the readers to concretize our proposed framework. We have updated the figure and specify in the caption where in the main text details the important concepts.
>
> -SoRB:
> We agree it is relevant and cite in the Related Work. In principal, their approach could be applied to control the low-level Worker behavior in our hierarchical setup, but it is unclear how their method would adapt to the stochastic elements of the tasks we study. Our work differs by focusing on identifying a single, generically-useful graph that can be applied to many tasks under a persistent environment. In contrast, SoRB builds a graph on the fly based on an internal distance estimator and the set of states available in the replay buffer.
>
> -Waypoint selection guarantee:
> There is no theoretical guarantee. The graph is meant to provide a convenient and task-agnostic abstraction of the structure and dynamics of the environment, which we argue provides a scaffold for rapid and structured exploration. Recovering action sequences from subsequences of states amounts to identifying those states that can be used to summarize some rollouts. By extension, we regard these states as the best subset for summarizing the environment’s structure. Table 3 demonstrates that a world graph using states identified in this manner is more useful than one constructed around randomly chosen states. As such, while our motivation does not come with guarantees, we present significant empirical validation.
>
> -Go-Explore:
> Thank you and we have updated the related works. Our waypoints share similar spirits with their “cells” but are automatically learned by the binary latent model instead of using heuristics specifically defined for different tasks as in Go-Explore.
>
> -Explore during VAE training:
> We attempt to ensure that the starting points of exploration rollouts (which are used to train the VAE) expand as training progresses. For our testbed environment, we achieve this by having the agent navigate to one of the current iteration’s waypoint states before collecting the exploration rollout. Feeding the reconstruction error as a curiosity bonus also addresses this potential issue by encouraging the exploration policy to produce trajectories that confuse the VAE.
>
> -$\mu_0$
> The derivation of equation (2), where $\mu_0$ appears, is detailed in Appendix A, equation (8)-(11). We have updated the main text to make the interpretation of this term clearer.
>
> -Coverage of waypoints:
> Our learned waypoints are visualized in Appendix D Figure 5, which have an even and comprehensive coverage over the environment. During graph learning, action reconstruction rate is our criterion in determining the sufficiency of exploration. The follow-up RL experiments further validate the coverage of the learned graph is sufficient to improve learning on a variety of downstream tasks.  One future direction, although beyond the scope of this work, is to record a buffer of areas concerning trajectories with high reconstruction errors during VAE training and focus exploration more on those areas.
>
> -Curiosity with RL baselines:
> No, this intrinsic reward is only used for the task-agnostic VAE phase (i.e. for learning the graph). A portion of the baseline Feudal Network is initialized from the goal-conditioned policy trained while learning the VAE, so it inherits some of the behavior shaped by the intrinsic reward.
>
> -Samples used by VAE training:
> We have updated the main text and brought forward the training information regarding sample complexity from Appendix C. In comparison to the RL stage, the amount of agent-environment interactions are much more efficient and the learning results are reused for all different downstream tasks in the RL stage.
>
> -Graph update during policy training and off-policy on graph training rollouts:
> In theory, updating graph in midst of policy training is possible and it can be an interesting future direction. Using VAE training rollouts as off-policy data is not straightforward in many commonly occurred cases. For example, for the Door-Key task, the data collected when training the VAE would not include the door or key.
>
> -Table 3
> We reference Table 3 in “Benefits of Learned Waypoints” Section 5.1, where we compare our results to results obtained using graphs with randomly chosen waypoint states. We have updated the table legend to make the meaning of the presented values clearer.

---

### Decision · Program_Chairs · 2019-12-19

**Decision:**

Reject

**Comment:**

This paper introduces an approach for structured exploration based on graph-based representations.  While a number of the ideas in the paper are quite interesting and relevant to the ICLR community, the reviewers were generally in agreement about several concerns, which were discussed after the author response. These concerns include the ad-hoc nature of the approach, the limited technical novelty, and the difficulty of the experimental domains (and whether the approach could be applied to a more general class of challenging long-horizon problems such as those in prior works). Overall, the paper is not quite ready for publication at ICLR.